# EyeWeS: Weakly Supervised Pre-Trained Convolutional Neural Networks for Diabetic Retinopathy Detection

**Pedro Costa**[1], **Teresa Araújo**[1,2], **Guilherme Aresta**[1,2], **Adrian Galdran**[1], **Ana Maria Mendonça**[2], **Asim Smailagic**[3], and **Aurélio Campilho**[2]

[1]INESC TEC
[2]Faculty of Engineering, University of Porto
[3]Department of Electrical and Computer Engineering, Carnegie Mellon University
[1]{pvcosta,tfaraujo,gmaresta,adrian.galdran}@inesctec.pt
[2]{amendon,campilho}@fe.up.pt
[3]asim@cs.cmu.edu

## Abstract

Diabetic Retinopathy (DR) is one of the leading causes of preventable blindness in the developed world. With the increasing number of diabetic patients there is a growing need of an automated system for DR detection. We propose EyeWeS, a general methodology that enables the conversion of any pre-trained convolutional neural network into a weakly-supervised model while at the same time achieving an increased performance and efficiency. Via EyeWeS, we are able to design a new family of methods that can not only automatically detect DR in eye fundus images, but also pinpoint the regions of the image that contain lesions, while being trained exclusively with image labels. EyeWeS improved the results of Inception V3 from $94.9\%$ Area Under the Receiver Operating Curve (AUC) to $95.8\%$ AUC while maintaining only approximately $5\%$ of the Inception V3's number of parameters. The same model is able to achieve $97.1\%$ AUC in a cross-dataset experiment. In the same cross-dataset experiment we also show that EyeWeS Inception V3 is effectively detecting microaneurysms and small hemorrhages as the indication of DR.

## 1 Introduction

Diabetic Retinopathy (DR) is a worldwide leading cause of preventable blindness, affecting more than $25\%$ [1] of the estimated 425 million diabetic patients in the world. The prevalence of diabetes is expected to grow to 629 million by 2045, and the number of patients requiring treatment will increase significantly in the following years [2]. In this context, early DR detection is important for successful treatment and thus large-scale screening programs are regularly implemented by hospitals and local authorities in both developed and developing countries. In these programs, diabetic patients are called to a clinic to acquire eye fundus images, which are then sent to ophthalmologists to perform the clinical diagnosis. Moreover, $34\%$ of diabetic patients live in rural areas [2], where the access to medical specialists and screening programs is scarce. Also, the large number of images to analyze and external factors such as stress due to high clinical work-loads hinder the diagnosis procedure. For these reasons, Computer-Aided Diagnosis (CAD) systems that are capable of detecting signs of DR are becoming increasingly important for screening the growing number of diabetic patients and reaching a larger percentage of this population.

International Conference on Medical Imaging with Deep Learning (MIDL 2018), Amsterdam, The Netherlands.

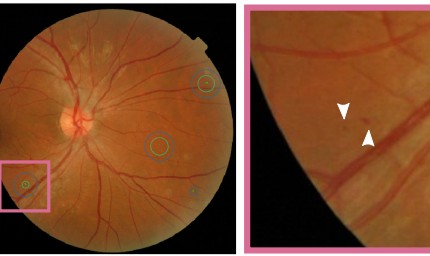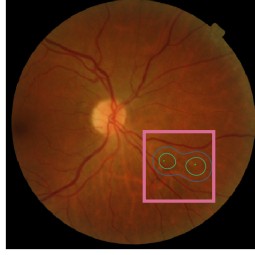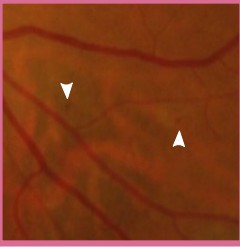

Figure 1: **EyeWeS is able to detect DR lesions being trained with image labels only.** The small green dots correspond to ground-truth segmentations of microaneurysms which are the indication of Diabetic Retinopathy. Given a new image, our method not only outputs whether an image displays signs of DR or not but also highlights the regions that contain lesions.

Over the last years, Deep Learning has become the standard approach for the development of CAD systems. However, the development and application of these systems in real practice is hindered by the lack of annotated data, which is expensive to obtain, as well as the lack of explainability of the predictions. To deal with these obstacles, in this paper, we introduce a novel general approach to train modern Convolutional Neural Network (CNN) architectures for the task of DR detection, illustrating how to easily convert such architectures into weakly-supervised models. This conversion removes the need of data annotated lesion-wise at the pixel-level for training, while maintaining the ability to pinpoint regions of the image that contain lesions relevant to diagnosis, as illustrated in Figure 1. The main contributions of this work are hence as follows:

- *Accuracy*: Our method is highly accurate, obtaining 95.8% AUC in Messidor[3] and 97.1% AUC in E-ophtha MA[4] in a cross-dataset experiment.

- *Explainability*: The designed approach finds regions with signs of DR. The model detects relevant microaneurysms for diagnosis while being trained with image labels only.

- *Efficiency*: The approach introduced here represents a straightforward technique to convert any current CNN architecture designed for the task of classification into a weakly-supervised model with a much reduced set of parameters to be learned.

- *Speed*: Our method is fast to both classify and explain the results, taking only $38.92 \pm 2.67$ milliseconds per image on a laptop's mobile Nvidia GTX 1060 GPU.

The remaining of this work is organized as follows. In the next section, a concise overview of deep learning for medical imaging is provided. We also introduce the notions of Transfer Learning and Weakly-Supervised Learning for mitigating the large need of data in modern deep learning-based CAD systems. Next, we introduce our proposed methodology to simultaneously leverage the advantages of both techniques for the problem of DR detection on retinal images. We then provide a comprehensive evaluation of our approach in different public datasets. The paper is concluded with some discussion on the obtained results and potential future work.

## 2   Related Work

### 2.1   Deep Learning for Medical Image Analysis

In recent years, Deep Neural Networks (DNNs) have started to have a strong presence on the field of medical data analysis [5, 6]. In the context of fully-supervised learning (when annotated data is available), neural networks composed of relatively many stacked layers are designed so that the provided raw data is transformed through subsequent non-linear operations into a suitable output (disease presence, pixel class, or any other quantity of interest). During training, the model iteratively updates the weights/parameters within its hidden layers, eventually finding optimal hierarchical data representations for the problem. These learned representations are built relying only on the minimization of a suitable loss function, thus by-passing the need of manual feature engineering. However, this process needs large quantities of data and suitable regularization techniques in order to converge to proper solutions that can later generalize on unseen data.

A particularly successful kind of DNN are CNNs, which take advantage of the strong spatial correlation of visual data by learning a set of shared weights in the form of filters. Since the success of the AlexNet architecture on the 2012 ImageNet challenge [7], new models have regularly appeared in the literature with steadily improving performance. From VGG [8] to Inception [9] or Residual Networks [10], the general trend has been increasing the depth of the models in order to build more expressive architectures that can learn richer data representations.

In more recent years, medical image analysis applications have also strongly benefited from these advances in Computer Vision. Initial applications comprise two-dimensional image classification [11], detection [12] or segmentation [13] problems, but the technology is quickly extending to more complicated visual data modalities, spanning from three-dimensional MRI [14] or CT scan analysis [15] to medical video understanding. It is worth noting that, despite the success of recent CNN architectures [16, 17], notable architecture choices for CNN-based medical image analysis involving classification tasks on two-dimensional data still include standard models such as VGG, although recent large-scale medical applications [11, 18] seem to favor Inception v3 [9].

## 2.2 Transfer Learning for Medical Image Analysis

Learning suitable weights for the class of deep CNNs mentioned above can be hard in situations where there is few annotated data, as is typically the case in medical imaging. To overcome this issue, a viable alternative is to use pre-trained models on large datasets of natural images, thereby reducing the learning burden [19]. This is process is commonly referred to as Transfer Learning.

In Transfer Learning, the weights of a CNN are initialized from successful solutions on large datasets instead of being randomly generated. In this case, the hierarchical data representations learned by the network to solve a given task are assumed to be useful for a new, possibly unrelated, one. Typically, the transferred weights are often more suitable on initial layers of a model, since these tend to learn more general data abstractions, than on later layers, which learn more task-specific representations. Hence, when applying Transfer Learning in a visual domain that is substantially different from natural images, or when data is scarce, a common strategy consists of re-training with a higher learning rate layers of the model closer to its output, while only slightly fine-tuning base layers [19].

## 2.3 Weakly-Supervised Learning and Multiple Instance Learning

Another suitable solution for the lack of large data quantities are Weakly-Supervised methods. As opposed to Fully-Supervised methods, with Weak Supervision a model is trained to learn low-level information from data even if the corresponding low-level ground-truth is not available, but rather a higher-level source of information is present.

A form of Weak Supervision that has been particularly successful in biomedical applications is the Multiple Instance Learning (MIL) framework [20]. In MIL, instead of supplying a learning algorithm with pairs of inputs and labels, these labels are associated with sets of inputs, also known as *bags*. In the case of binary labels, the fundamental MIL assumption states that a positive bag (*e.g.* an image with signs of disease) contains at least one positive instance (*e.g.* a lesion).

Training MIL-based models requires only weak annotations. This is translated in practice to using only image-level labels, while still classifying images based on instance-level information, *i.e.* on the presence of lesions. Requiring weak annotations simplifies enormously data collection, which is a major issue in medical image analysis. However, these weaker annotations need to be compensated by larger datasets [20]. Accordingly, the availability of such large datasets is a typical prerequisite for training MIL-based algorithms. This need is emphasized if the models to be trained are deep CNNs, which contain a large number of parameters to be learned. For this reason, in this paper we propose to embed standard Transfer Learning strategies into a MIL framework. The technical details on how to achieve this goal are explained in the next section.

## 3 EyeWeS for explainable Diabetic Retinopathy detection

The approach proposed in this paper, referred to as EyeWeS, consists of a combination of MIL and Transfer Learning for deep CNNs. As such, the method is capable of formulating a decision regarding the presence of DR on a retinal image, while detecting the image's regions that better explain such

decision. We start by formalizing the problem through the MIL framework to gain some intuition on the problem. Then, following these insights, we outline a strategy to modify current state-of-the-art CNN architectures to better suit the *Standard MIL Assumption*.

## 3.1 Intuition

In order to train a MIL-based model on retinal images to detect DR based on lesion presence, our goal is to train an *instance* classifier from *bag* labels. In this context, an image is regarded as a *bag* composed of several rectangular spatial neighborhoods, *i.e.* image patches. These patches are considered as instances. Hence, the goal becomes to train a patch classifier from image labels only.

Since patch-level labels are not available, we consider these instance's labels $y_i$ as latent variables, because they are unknown during training. The latent variables can be combined by means of a pooling function $f$ into the (available) corresponding bag label $Y = f(y_1, ..., y_N)$. The pooling function is responsible of encoding the relation between the instances' and bag's labels.

The objective is to learn an instance classifier $P(y_i|x_i, \theta)$, where $x_i$ denotes the $i$-th instance in an image, and $\theta$ are the classifier's parameters. As the instances' labels are unknown, we can only maximize the likelihood $P(Y|y_1, ..., y_N, x_1, ..., x_N, \theta)$. Furthermore, we assume that $\{X, \theta\}$ and $Y$ are conditionally independent given $y$. This means that, given the labels of the instances, both the instances and the model's parameters do not provide information on the likelihood of $Y$. Therefore, the goal is to find the parameters $\theta$ that maximize the likelihood:

$$\theta = \arg\max_{\theta} P(Y|y_1, ..., y_N; \theta) \tag{1}$$

There are some design choices that need to be made before implementing this idea, namely: 1) what is the learning algorithm to use for the patch classifier and 2) what pooling function $f$ to use.

## 3.2 Fully-Convolutional Patch Classifier

We choose to use a CNN for the patch classifier algorithm. However, instead of training a CNN on individual image patches and their corresponding labels, we use a Fully-Convolutional Network (FCN) architecture, which is capable of classifying image patches trained only with the full images and the respective labels. For that, we build on receptive field properties of Convolutional Layers.

The receptive field can be defined as the local region in the input space that affects the value of a particular activation unit in a given layer. For instance, the output of a $3 \times 3$ Convolutional Layer has a $3 \times 3$ receptive field, while two consecutive $3 \times 3$ Convolutional Layers have a $5 \times 5$ receptive field as depicted in Figure 2. Therefore, we can design a Fully-Convolutional Network architecture such that the receptive field of the last layer's units corresponds to the desired patch size. Moreover, a FCN architecture leads to higher efficiency when there is overlap between the image patches, since the features extracted on the overlapping region are only computed once, while in a patch-based CNN these would be extracted independently for each patch.

## 3.3 Combining Patch Predictions

The computed image patch predictions are then combined into the image label using a pooling function $f$. In EyeWeS, the *Standard MIL Assumption* described above is followed. This assumption can be implemented with the max-pooling function $Y = max(y_1, ..., y_N)$, since a single positive instance is enough to predict the full image as positive. As the $max$ function is almost everywhere differentiable, a misclassification loss function can be directly applied to $Y$ and the network trained with the backpropagation algorithm.

It is worth noting that, in practice, most binary disease detection problems follow the *Standard MIL Assumption* since the presence of a disease can usually be inferred by the existence of a lesion in the image. As lesions typically have well known average dimensions, the patch size can be defined based on this medical knowledge. However, since the prediction is based on local information, EyeWeS can not be used for classifying the severity of a disease, since this usually requires the detection, classification and counting of multiple lesions that can be spread over the entire image. This breaks the *Standard MIL Assumption* as it is necessary to combine information from all patches and, consequently, a different pooling function would be required.

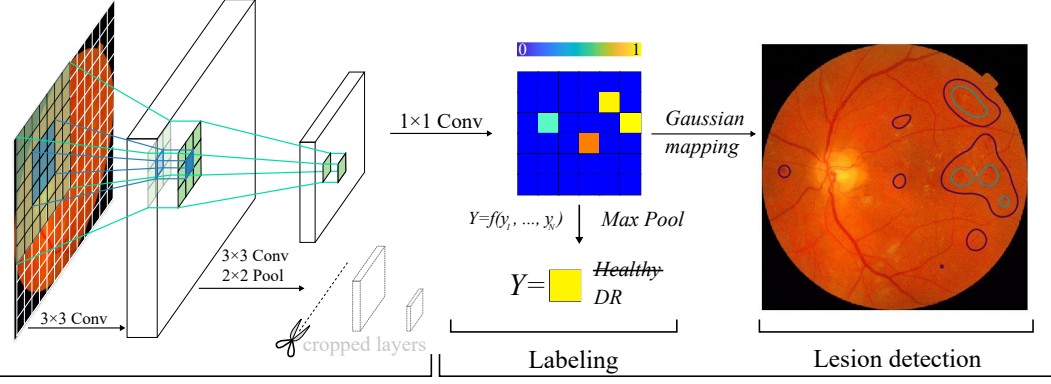

Figure 2: Early layers of pre-trained networks are kept to constrain the receptive field of their output units. For instance, the receptive field of the first $3 \times 3$ Convolution is a $3 \times 3$ region of the input image, but the receptive field of two $3 \times 3$ Convolutions is a $5 \times 5$ region of the input. Then, $1 \times 1$ Convolution layers are used to classify the input patches. Finally, the image prediction is obtained by max pooling all the patch predictions.

### 3.4   Transfer Learning in EyeWeS

When few training data is available, training a deep CNN from scratch can be unfeasible. In these situations, it is common practice to initialize the network from pre-trained weights and finetune it for a given new task. However, standard CNN architectures have been designed to classify the entire image and thus the receptive field of the last layer is of the same size as the input, which is not useful for patch-level classification.

In EyeWeS, we propose to find the layer of the pre-trained network which receptive field is closer to a desired patch size and discard all subsequent layers. This patch size should be roughly large enough to contain the size of the lesions to be detected within the image. The output of this intermediate layer is then a $H \times W \times K$ tensor with a $K$-dimensional feature vector for each of the $N = H \times W$ image patches. Then, $1 \times 1$ Convolutional Layers are added in order to perform patch classification without increasing the receptive field, as illustrated in Figure 2.

It should also be considered that standard pre-trained models are typically trained on low-resolution images. For instance, CNNs trained on ImageNet usually require $224 \times 224$ pixel images as input. However, some lesions in medical images, such as retinal microaneurysms, are too small to be detected at such low resolutions. In our case, since EyeWeS actually classifies patches from input images, the full image resolution is not relevant because the necessary information is locally present in each image patch. In this work, we apply models pre-trained on ImageNet on images with more than twice the resolution of the images on which the original models were trained.

## 4   Application and Results

We train and test EyeWeS on the Messidor [3] dataset for solving the task of DR detection. Messidor is a publicly available dataset with 1200 images of different resolutions: 1440×960, 2240×1488 and 2304×1536 pixels. The images were acquired by three opthalmologic departments using a non-mydriatic retinograph and have a field-of-view of 45°. Images were annotated by specialists with the corresponding DR grade, *i.e.*, DR level of severity, ranging from 0 (no pathology) to 3 (most severe stage). These stages are determined according to the number of lesions in the image: microaneurysms, hemorrhages and neovascularizations. Grade 0 images are characterized by containing no lesions, whereas grade 1 images are characterized by the presenece of up to five microaneurysms and no hemorraghes, since microaneurysms are the earliest sign of DR. Grade 2 and 3 images contain a greater number of lesions. In Messidor, only image-level labels are provided and no lesion-level information is available.

Table 1: **EyeWeS' receptive field size.** The optimal receptive field size in pixels (px) for each network is much smaller than the original image resolution. The number of patches being classified $N$ is large due to the large patch overlap, providing a good spatial lesion detection resolution.

| Architecture | Receptive Field (px) | Overlap (px) | $N$ |
|---|---|---|---|
| EyeWeS VGG16 (block4_conv1) | $52 \times 52$ | $44 \times 44$ | $64 \times 64$ |
| EyeWeS ResNet50 (add 3) | $30 \times 30$ | $26 \times 26$ | $127 \times 127$ |
| **EyeWeS Inception V3 (mixed 2)** | $114 \times 114$ | $106 \times 106$ | $61 \times 61$ |

In this work we are only interested in detecting DR and, therefore, we pose the problem as binary one-vs-all classification task, distinguishing between healthy images (*i.e.* grade 0) and DR images (*i.e.* grades 1, 2 and 3). We randomly divided Messidor into three sets: a training set with 768 images (64%), a validation set with 192 images (16%), and a test set with the remaining 240 images (20%).

We also perform a cross-dataset experiment to both evaluate the generalization capabilities of EyeWeS and perform a qualitative assessment of its lesion detection capability. For this, we used the E-ophtha MA dataset [4], which contains 148 images with microaneurysms or small hemorrhages (together with pixel-level segmentations) and 233 images with no lesions.

### 4.1  Implementation Details

The training process for EyeWeS proceeds in 3 steps: 1) select the layer of the given pre-trained model to use; 2) train the newly added layers, while keeping the parameters of the pre-trained layers constant and 3) train the full model. All our experiments were implemented using Keras [23] and all networks were pre-trained on ImageNet.

In this work, we experimented with three different CNN architectures: VGG16 [8], Inception V3 [9] and Resnet50 [10]. The number of layers of the given pre-trained model to reuse was treated as a hyper-parameter and it was chosen independently for each architecture using grid-search over all intermediate layer blocks. Following Keras' naming conventions, the layers that achieved optimal performance for each architecture were: block4_conv1 (VGG16), mixed 2 (Inception V3) and add 3 (Resnet50). Details on the receptive field size and overlap of each architecture are displayed in Table 1 along with the number of patches that are classified prior to the max pooling operation.

Two $1 \times 1$ Convolution Layers are used after the output of the selected intermediate layer, the first one with 1024 units, followed by a LeakyReLU activation function with slope of 0.02 and the second one with a single unit, followed by a sigmoid activation function in order to perform patch-level classification. These last two Convolution Layers are trained for the first 30 epochs with a learning rate of $10^{-3}$. Then, the full model is trained with a learning rate of $2 \times 10^{-4}$ using early stopping with a patience of 15. The Adam [24] optimizer was used in both steps with default parameters.

The input images were cropped to the Field-of-View and resized to $512 \times 512$ pixels. In order to improve the generalization of the model, dataset augmentation was used. On top of the standard horizontal and vertical flip, translation, rotation and scaling of the input images, a color balance method [25] was also used as a dataset augmentation operator as it was shown to improve results in segmenting vessels in eye fundus images.

Table 2: **EyeWeS' DR detection results on Messidor**. EyeWeS achieves state-of-the-art Area Under the Receiver Operating Curve (AUC) compared with other weakly-supervised methods.

| Architecture | AUC | Weakly-Supervised |
|---|---|---|
| Costa *et al.* [21] | 90.00% | ✓ |
| Zoom-In-Net [22] | 92.10% | ✓ |
| VGG16 | 83.44% | |
| ResNet50 | 93.77% | |
| Inception V3 | 94.97% | |
| EyeWeS VGG16 | 90.00% | ✓ |
| EyeWeS ResNet50 | 94.53% | ✓ |
| **EyeWeS Inception V3** | **95.85%** | ✓ |

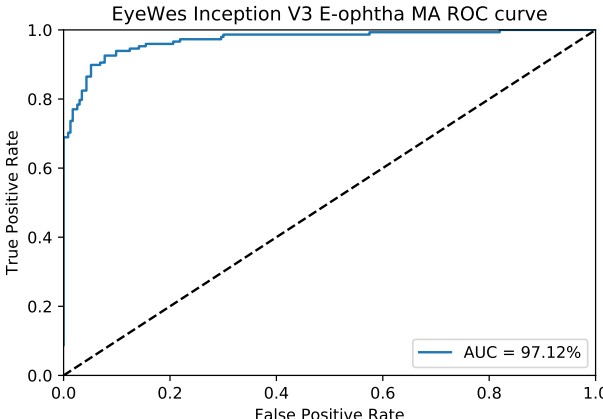

Figure 3: **EyeWeS generalizes to other datasets**. Receiver Operating Curve of EyeWeS with Inception V3 on the E-ophtha MA dataset.

After running EyeWeS, we obtain an array of patch predictions and an image-level prediction stating its probability of being affected by DR. The patch-wise predictions allow to generate an attention map highlighting the most relevant diagnosis regions. For that, the patch predicions' array needs to be upsampled to the image resolution. In this process, each patch only contributes to a given pixel value if that pixel is inside the patch's receptive field. Each pixel value is then obtained by a weighted average, where the weights are drawn from a Normal distribution with the mean located at the patch's center and standard deviation corresponding to the network's receptive field.

## 4.2   Quantitative Results

In order to properly evaluate our method, we performed two experiments: 1) we tested on Messidor and 2) we performed a cross-dataset experiment, testing EyeWeS on E-ophta MA. Results from experiment 1) are shown in Table 2. The Inception V3 network obtains the best results ($95.85\%$ AUC) closely followed by ResNet50 ($94.53\%$). On the other hand, VGG16 is not able to achieve comparable results, attaining only $90\%$ AUC. Nonetheless, it is worth noting that this result is already at the same performance level as other recent works [21, 22].

These results indicate that EyeWeS can achieve state-of-the-art performance even when training on medium-sized datasets (in this case, 768 images were used for training). This is particularly relevant for the problem of DR detection: in [22], it was mentioned that Messidor was too small to properly train a Deep CNN, and the method was trained with the help of external data. In our case, thanks to the use of pre-trained networks, EyeWeS can be trained without overfitting the available data. This hints to our method being more robust to the dataset size, due to EyeWeS focusing on patches of the image instead of on the entire image.

We also compared EyeWeS with standard pre-trained architectures. For that, we removed the last layer of each given network and applied a Global Average Pooling layer to accommodate for the increase in image resolution. Then, similarly to the EyeWeS, we added two $1 \times 1$ Convolution layers, the first one with $1024$ units followed by a LeakyReLU activation function and the last one with a single unit followed by a sigmoid. All networks were trained in the same conditions as EyeWeS. As seen in Table 2, EyeWeS consistently obtains better AUC results than their standard counterparts.

Finally, in order to fairly evaluate if EyeWeS generalizes to other datasets, we selected the model that performed best in Messidor, in this case, the EyeWeS Inception V3 model, and tested it on the full E-ophta MA dataset. We obtained $97.12\%$ AUC in this cross-dataset experiment, as shown in Figure 3, indicating that our method is effectively finding microaneurysms or small hemorrhages.

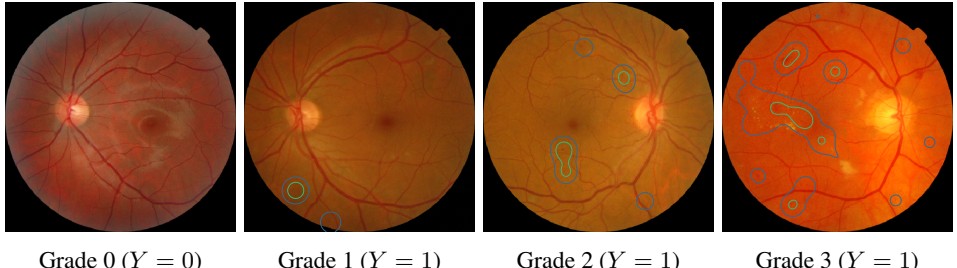

| Grade 0 ($Y = 0$) | Grade 1 ($Y = 1$) | Grade 2 ($Y = 1$) | Grade 3 ($Y = 1$) |

Figure 4: **Examples of EyeWeS's explainable results on Messidor's test images.** All images were correctly labeled by EyeWeS. Grades are displayed for visualization purposes only, we do not use grade information nor lesion segmentations when training our method. Attention map: 0 ▭ 1

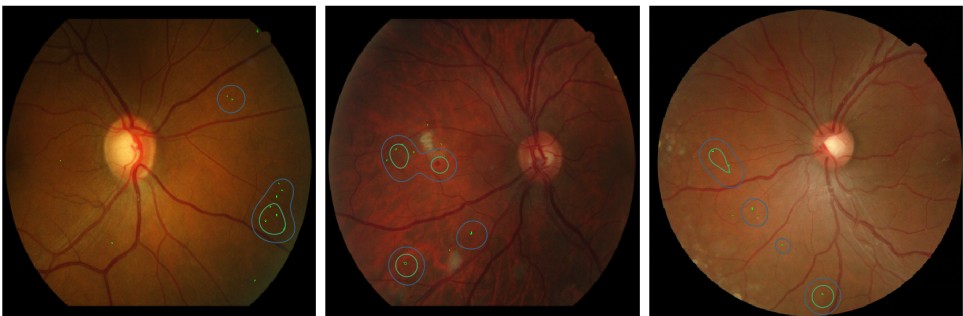

Figure 5: **EyeWeS detects microaneurysms**. Microaneurysm true locations from E-ophtha MA are displayed in light green. It is possible to see that the method's detection coincides with the dataset's true locations. There are still some false negatives, especially on smaller lesions. Attention map: 0 ▭ 1

## 4.3   Explainability

EyeWeS produces patch-level predictions even though it is only trained with image labels, which can help in visually interpreting the reasons of the models' decisions, enhancing its interpretability. In this section we qualitatively evaluate the patch-level predictions both on Messidor and E-ophtha.

From the patch-level predictions, attention maps were built following the method outlined at the beginning of this section. As expected, the attention maps produced by EyeWeS focus on eye lesions. In Figure 4 it is possible to visualize some sample results on images from Messidor's test set with different severity labels. As can be observed, regions of the image that the method considers pathological lie on top of red lesions.

It is also interesting to observe that, as EyeWeS max-pools the local patch predictions into a single global image prediction, the optimization process only promotes the finding of a single pathological patch per image. This realization could lead to the idea that such a method would have a large false negative rate. However, in practice we see that our method is able to find more than one lesion per image as seen in Figure 4.

Another interesting result of our method is that it focuses on small lesions while ignoring the larger more visible ones. It is possible to see in the grade 3 image of Figure 4 that EyeWeS does not focus on large hemorrhages and bright yellow lesions. These results potentially mean that the method was able to correctly identify microaneurysms as the earliest and most subtle indication of DR.

In order to test this last hypothesis, we visualize the attention maps produced by our method on E-ophtha MA in Figure 5. It is possible to see that the model's detections coincide with microaneurysm locations. The smaller lesions may be composed of as few as 3 pixels within a $512 \times 512$ pixel image. It can also be appreciated that in some cases, EyeWeS tends to miss these smaller lesions, which is an indication that the performance can still be improved by increasing the resolution of the input images.

Table 3: **EyeWeS is faster and has fewer parameters.** Mean inference time in milliseconds with 95% CI for a single image and number of parameters of each network.

|  | EyeWeS Time (ms) | EyeWeS Parameters | Full Time (ms) | Full Parameters |
|---|---|---|---|---|
| Inception V3 | $38.92 \pm 2.67$ | $1,240,673$ | $65.50 \pm 4.04$ | $23,715,265$ |
| Resnet50 | $33.74 \pm 1.30$ | $498,049$ | $81.93 \pm 3.74$ | $25,691,009$ |
| VGG16 | $57.08 \pm 1.25$ | $3,446,081$ | $75.39 \pm 1.78$ | $15,245,121$ |

### 4.4 Efficiency and Inference Time

In this section we explore the impact of our method in the reduction of the number of parameters and inference time. For the performance measurements, experiments were run on a laptop equipped with a mobile Nvidia GTX 1060 GPU. Inference times were averaged over 100 experiments. The resulting performances, within a 95% confidence interval (CI), are reported in Table 3. In addition to EyeWeS's version of each of the three considered baseline CNN's architectures, we show also for comparison the performance of the corresponding original versions. In this case, EyeWeS ResNet50 was $2.43\times$ faster than its original counterpart at test time, while EyeWeS Inception V3 was $1.68\times$ faster, and EyeWeS VGG16 was $1.32\times$ faster.

Implementing EyeWeS allows to reduce the number of parameters in more than a quarter with respect to the corresponding reference architectures, as can also be observed from Table 3. In the extreme case of Resnet50, the number of parameters was reduced to less than 2% of the original number. EyeWeS also reduces the number of parameters of Inception V3 to approximately 5% of its original number, while VGG16 parameter numbers are reduced to approximately 23%.

## 5 Conclusions

We have introduced EyeWeS, which addresses the problem of explainable detection of Diabetic Retinopathy from eye fundus images. Out of global image labels, our method trains a patch classifier that can be used at test time to detect the regions of the image that contain lesions of interest for DR diagnosis. The proposed method has been shown to be capable of accurately detecting DR on retinal images, making its decision based on local information. EyeWeS's architecture allows to pinpoint the spatial locations triggering the model's decisions, which enhances its explainability.

EyeWeS has been comprehensively validated through experimental tests on the Messidor dataset for DR detection, achieving state-of-the-art results. In addition, the same model has been tested on E-ophtha MA without further re-training. Comparison with the pixel-wise lesion ground-truth available for E-opththa MA showed that EyeWeS's patch classifier detects microaneurysms as small as 3 pixels in diameter. It has also been shown that it is possible to decrease the number of parameters by more than 98% with respect to the reference CNN architecture from which EyeWeS is derived and still obtain more accurate results with the added ability to explain the results.

In the future, EyeWeS will be trained on higher resolution retinal images, in order to better detect the smaller microaneurysms. In addition, the same ideas can be tested in other medical imaging problems and datasets, as long as the *Standard Multiple Instance Assumption* holds.

### Ackowledgments

This work is financed by the ERDF – European Regional Development Fund through the Operational Programme for Competitiveness and Internationalisation - COMPETE 2020 Programme, and by National Funds through the FCT - Fundação para a Ciência e a Tecnologia within project CMUP-ERI/TIC/0028/2014 and by the North Portugal Regional Operational Programme (NORTE 2020), under the PORTUGAL 2020 Partnership Agreement within the project "NanoSTIMA: Macro-to-Nano Human Sensing: Towards Integrated Multimodal Health Monitoring and Analytics/NORTE-01-0145-FEDER-000016". Teresa Araújo is funded by the FCT grant contract SFRH/BD/122365/2016. Guilherme Aresta is funded by the FCT grant contract SFRH/BD/120435/2016.

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
