# OpenReview forum: "EyeWeS: Weakly Supervised Pre-Trained Convolutional Neural Networks for Diabetic Retinopathy Detection"
_MIDL.amsterdam/2018/Conference — Submitted to MIDL 2018_

### Review · AnonReviewer2 · 2018-05-04
**The usage of the term weakly supervised is somehow misleading**

**Rating:** 2
**Confidence:** 2

**Review:**

The paper applies CNN architectures (VGG16, ResNet50, and Inception v3) to detect diabetic retinopathy. The CNN architecture is built in a way that allows to inspect pre-maxpooling layer and display the location of lesions. The method is evaluated in terms of classification AUC on two datasets: Messidor and E-optha MA.

I like the application of deep learning for DR detection and an emphasis on model explainability. However, I have some comments\concerns with respect to the paper:

1. The usage of the term weakly supervised is somehow misleading. It seems that the paper does weakly supervised localization of lesions; however, the emphasis of the evaluation is on showing the results for classification task. Since the focus of the paper seems to be weakly supervised localization the main part of experiment should justify the superiority of the method in terms of localization and the classification task should be only used to show that the method performs comparable to state of the art.
2. Given point #1, it seems that the focus of the paper should be either on weakly supervised lesion localization using pre-trained CNNs or on supervised lesion classification using pertained CNNs.
3. If the goal to the paper is to do DR classification, the method should be compared to [11] and some discussion of the result w.r.t. published methods should present in the paper.
4. It is not clear how the spatial map in figure 2 is computed. It seems like there is a sigmoind nonlinearity on each position of the feature map, this should be explained clearly in the paper.
5. Going back to point #1, the weakly-supervised tick in Table 2 is not clear. It seems like the table presents image classification results, thus, all the methods are supervised.
6. It would be beneficial to display number of parameters for each method in table 2. It is not clear why VGG16 would perform worse than EyeWeS VGG16. Is it because of different number of parameters? Or, due to sigmoid nonlinearity in pre-maxpooling feature map?
7. Table 3. VGG16 has 100+M parameters. What is the exact architecture the authors refer to with "full parameters VGG16" (with 15+M parameters)?
8. Eq. 1: Should we condition P(Y|y_1, ... y_n) also on x_i?
9. "This breaks the Standard MIL Assumption as it is necessary to combine information from all patches... ", It is not clear why it would be the case.
10. The related work section could be shortened, with more emphasis on concepts from semi-supervised learning, transfer learning, weakly supervised learning and less on general DL approaches.


**Special Issue:**

No

---

### Review · AnonReviewer1 · 2018-05-08
**Interesting technique for lesion localization**

**Rating:** 4
**Confidence:** 2

**Review:**

This is an interesting technique that I believe could be useful in the field.  Often medical images are only partially annotated and as medical datasets grow there is a need for method to partially automated the annotation of those datasets.  The paper is well written and includes many important components to a machine learning paper such as detailed descriptions of the hyper-parameters used in the training of the CNN.

There are two issues that the authors could resolve before presenting this work.

First the description of the CNN architecture combining the information in sections 3.4 and 4.1 is insufficient.  The reference to the number of layers use and the final pretrained part of the architecture should be more fully described, or possibly a schematic of the final network architecture provided.

Finally as this paper is presented as primarily intended to provide a method for lesion localization and the authors are using the E-ophtha dataset and FROC curve should be provided for that dataset along with some comparison to the existing literature on lesion detection using the same data.  They do not need to show state of the art performance compared to fully supervised methods, but providing that reference would be useful for evaluation purposes.

**Special Issue:**

No

---

### Review · AnonReviewer3 · 2018-05-17

**Rating:** 2
**Confidence:** 3

**Review:**



**Special Issue:**

No

---

### Decision · Program_Chairs · 2018-05-15
**Paper54 Acceptance Decision**

Reject